# Computational Modeling Insights into Extreme Heterogeneity in COVID-19 Nasal Swab Data

**DOI:** 10.3390/v16010069

**Published:** 2023-12-30

**Authors:** Leyi Zhang, Han Cao, Karen Medlin, Jason Pearson, Andreas C. Aristotelous, Alexander Chen, Timothy Wessler, M. Gregory Forest

**Affiliations:** 1Department of Mathematics and Carolina Center for Interdisciplinary Applied Mathematics, University of North Carolina at Chapel Hill, Chapel Hill, NC 27599, USA; 2Simulations Plus, Inc., 6 Davis Dr., Durham, NC 27709, USA; 3Department of Mathematics, The University of Akron, Akron, OH 44325, USA; 4Department of Mathematics, California State University, Dominguez Hills, CA 90747, USA; 5Department of Applied Mathematics, University of Colorado at Boulder, Boulder, CO 80309, USA; 6Departments of Applied Physical Sciences and Biomedical Engineering, University of North Carolina at Chapel Hill, Chapel Hill, NC 27599, USA

**Keywords:** SARS-CoV-2, nasal infection, sensitivity analysis, extreme outcome heterogeneity

## Abstract

Throughout the COVID-19 pandemic, an unprecedented level of clinical nasal swab data from around the globe has been collected and shared. Positive tests have consistently revealed viral titers spanning six orders of magnitude! An open question is whether such extreme population heterogeneity is unique to SARS-CoV-2 or possibly generic to viral respiratory infections. To probe this question, we turn to the computational modeling of nasal tract infections. Employing a physiologically faithful, spatially resolved, stochastic model of respiratory tract infection, we explore the statistical distribution of human nasal infections in the immediate 48 h of infection. The spread, or heterogeneity, of the distribution derives from variations in factors within the model that are unique to the infected host, infectious variant, and timing of the test. Hypothetical factors include: (1) reported physiological differences between infected individuals (nasal mucus thickness and clearance velocity); (2) differences in the kinetics of infection, replication, and shedding of viral RNA copies arising from the unique interactions between the host and viral variant; and (3) differences in the time between initial cell infection and the clinical test. Since positive clinical tests are often pre-symptomatic and independent of prior infection or vaccination status, in the model we assume immune evasion throughout the immediate 48 h of infection. Model simulations generate the mean statistical outcomes of total shed viral load and infected cells throughout 48 h for each “virtual individual”, which we define as each fixed set of model parameters (1) and (2) above. The “virtual population” and the statistical distribution of outcomes over the population are defined by collecting clinically and experimentally guided ranges for the full set of model parameters (1) and (2). This establishes a model-generated “virtual population database” of nasal viral titers throughout the initial 48 h of infection of every individual, which we then compare with clinical swab test data. Support for model efficacy comes from the sampling of infection dynamics over the virtual population database, which reproduces the six-order-of-magnitude clinical population heterogeneity. However, the goal of this study is to answer a deeper biological and clinical question. *What is the impact on the dynamics of early nasal infection due to each individual physiological feature or virus–cell kinetic mechanism?* To answer this question, global data analysis methods are applied to the virtual population database that sample across the entire database and de-correlate (i.e., isolate) the dynamic infection outcome sensitivities of each model parameter. These methods predict the dominant, indeed exponential, driver of population heterogeneity in dynamic infection outcomes is the latency time of infected cells (from the moment of infection until onset of viral RNA shedding). The shedding rate of the viral RNA of infected cells in the shedding phase is a strong, but not exponential, driver of infection. Furthermore, the unknown timing of the nasal swab test relative to the onset of infection is an equally dominant contributor to extreme population heterogeneity in clinical test data since infectious viral loads grow from undetectable levels to more than six orders of magnitude within 48 h.

## 1. Introduction

One silver lining of the COVID-19 pandemic is the unprecedented, global sharing of clinical and scientific data. These shared databases have revealed many insights into novel coronaviruses, and SARS-CoV-2 in particular, including the astounding number and speed of protein mutations. At the same time, many open questions have been exposed in the cell biology of respiratory viral infections. One particular open question centers on the mechanisms affected by the SARS-CoV-2 protein mutations and their impact on onset and progression of infection, including whether the impacts are uniform versus heterogeneous in the population. This causal, mechanistic link between viral RNA modifications and human respiratory infection outcomes is extremely cloudy as there are many complex processes that lie between the molecular and organ scales. In this article and study, we focus on one remarkable aspect of COVID-19 clinical data. Namely, nasal swab titers collected from individual, non-hospitalized, positive tests have varied by six orders of magnitude [1,2,3,4,5,6,7,8,9]. This dramatic heterogeneity has persisted throughout the pandemic and therefore within and between variants, in all countries reporting data, and prior to and after previous SARS-CoV-2 exposure, infection, or vaccination. Due to the unprecedented global sharing of clinical data for COVID-19, it remains unclear whether this dramatic population heterogeneity in nasal infection is unique to SARS-CoV-2 or potentially generic to respiratory viruses. We turn to computational modeling to seek insights into the possible drivers of such dramatic heterogeneity in nasal infection tests between individuals.

In January of 2020, our group began development of a within-host, agent-based, computational model of human respiratory tract (RT) exposure to and infection by a novel virus. Like all models, choices must be made as to what features to include or not, and the efficacy of model predictions comes with the caveats of the choices made. For biologists and clinicians, as well as the practitioners of computational modeling, one should bear in mind the famous quote from 1976 by statistician George Box: “all models are wrong, some are useful”. In January of 2020, a physiologically faithful, spatially resolved model of the inhalation of a virus onto the air-mucus or air-alveolar liquid interface, and the subsequent diverse outcomes, did not exist. We built such a model, choosing to incorporate the kinetic processes of viral diffusion, virus-cell encounters, and, once a cell is infected, the processes of cellular uptake of the virus and viral hijacking of cellular machinery to make viral RNA copies, followed by cellular shedding of viral RNA copies into the airway surface liquid until cell death. Our group was well-positioned to build such a model because of (1) 25 years of research on lung physiology and biology, in particular the lung branching structure with generation-dependent mucus layer thickness and clearance velocity toward the trachea due to propulsion by beating cilia and (2) 10 years of research on sexually transmitted viral infections in the female cervicovaginal tract, which is also coated with a mucus layer that drains gravitationally. The baseline model [10] incorporates the complex anatomy and physiology of the human RT, as well as the kinetic processes of virion diffusivity $D_{v}$ in the mucosal barrier, the probability $p_{\mathrm{infect}}$ of cell infection per virion encounter, the latency time $t_{\mathrm{latency}}$ of an infected cell prior to shedding viral RNA, and, once shedding starts, the shedding rate $r_{\mathrm{shedding}}$ of infectious viral RNA copies until cell death. The latency time $t_{\mathrm{latency}}$ spans the moment of cell infection until the onset of extracellular shedding of viral RNA.

We note that these “model features and mechanisms” are examples of the choices that one must make in order to capture, in an approximate manner, sufficient key impacts on outcomes. For example, (1) we assume the mucus layer is uniformly thick in each generation and moves like an escalator with the same velocity at each height of the layer; (2) we assume a virus, when it diffuses through the airway surface liquid to encounter an infectable cell (assumed to be  50% of epithelial cells), infects or not according to a flip of a biased coin (e.g., infection 1 out of 5 encounters) based on best available experimental data; and (3) once a cell is infected, we impose a latency time (a prescribed delay phase) after which the cell begins shedding viral RNA copies at some prescribed rate, but we do not resolve the processes and timescales for cellular uptake of the virus and hijacking of cellular machinery to produce viral RNA copies. There is strong cell culture evidence linking protein mutations and the pathway and speed for cellular uptake of the virus. Since infected cells typically live longer than 2 days, cell death does not enter the present study. All of the above mechanistic parameters and physiological features were estimated at mean population values in [10], providing a framework to simulate outcomes of human respiratory infection that is physiologically faithful and incorporates the diffusive mobility of viruses in airway surface liquids and the kinetics of virus–cell infection, replication, and shedding. Below, we summarize the mathematical structure of this model. Additional extensions of the model to include innate [11] and adaptive [12] immunity have been developed, but they are not included in this study motivated by the overwhelming evidence of immune escape over the 48 h or longer post-infection period [9,13,14,15,16,17,18,19,20,21], independent of vaccination status and prior infection.

One important advantage of computational modeling in biology is that, despite the assumptions that render the model only an approximation of in vivo behavior, the model is able to provide predictions of outcomes and test whether features or mechanisms within the model are sufficient to replicate clinical or experimental observations and thus pose candidates for experimental or clinical confirmation. Indeed, the model may shed insights into the relative importance of physiological and in vivo conditions underlying clinical data, as well as the relative importance of ex vivo experimental controls underlying experimental data. We note two such illustrations in our previous work, which further motivate the present study. In [10], in the nasal passage, trachea, and the first few upper branches of the human RT, mucus layer advection is strong and dominates diffusion of viruses while in the mucus layer. In vivo, strong mucus advection creates, from each initial infected cell, “thin streaks” of infected epithelial cells and shed viruses within mucus. Further, mucus advection accelerates growth in viral load and infected cells relative to a stationary mucus layer of the same thickness that might exist in an ex vivo culture experiment. The upshot is that ex vivo cultures with identical mucus layers produce extreme underestimates of in vivo viral load and infected cells. In [22], using the same model and code from [10], we performed a limited parameter sensitivity study of viral load and infected cells in the nasal passage, e.g., by varying the kinetic parameters governing cell infection, replication, and shedding over ranges guided by the literature. The study was limited in that only kinetic parameters and ranges were considered, not physiological parameters, and further limited by the parameter search. Namely, each parameter variation was studied by fixing all other parameters at mean population estimates, and not sampling in all directions of the full parameter space. One can think of this sampling of parameter space as extremely sparse, with each search starting from the mean of all parameters and exploring one parameter direction at a time from the global parameter mean. In addition to an extremely limited sampling of parameter space, moving only one parameter at a time rather than the freedom to move along any direction in the parameter space, the search is blind to correlations between parameters and the physiology or mechanisms they represent. Nonetheless, the following results in [22] are suggestive and guide the present study.

First, it was discovered that model outcomes of viral load and infected cell count are extremely robust/insensitive to variations in $p_{\mathrm{infect}}$, and in fact negatively correlated with $p_{\mathrm{infect}}$. (This result suggests that spike mutations leading to stronger binding to cell receptors may very well increase the likelihood of infection from exposure but is *not* responsible for increased viral titers or infected cell counts.) As a consequence, to limit the dimension of parameter space we need to explore in this study, we fix $p_{\mathrm{infect}}=0.2$. Second, model outcomes are sensitive and positively correlated with $r_{\mathrm{shedding}}$. Therefore, since the experimental and clinical data on the replication rate of *infectious* RNA copies (virions) remain poorly understood, we allow for two decades of $r_{\mathrm{shedding}}$, 10–1000 infectious virions per day by infected cells in the shedding phase. Finally, model outcomes are found to be *exponentially sensitive* to linear variations in $t_{\mathrm{latency}}$. Therefore, based on prior [23,24] and continued [25] single-cell experimental resolution data, we explore $t_{\mathrm{latency}}$ spanning 3–9 h. (N.B. Since we fix $p_{\mathrm{infect}}=0.2$ in this analysis, results from [22] are presented in the Supplementary Materials to illustrate the remarkable robustness of outcomes to an order of magnitude variability in $p_{\mathrm{infect}}$.) Upper and lower bounds on all parameters, both physiological and in virus–cell infection kinetics, continue to be updated during the pandemic. Remarkably, none of the three cellular kinetic parameters in our model have been experimentally quantified. Therefore, we retain bounds on the sensitive parameters $r_{\mathrm{shedding}}$ and $t_{\mathrm{latency}}$ that are consistent with the literature noted above and fix the robust kinetic parameter $p_{\mathrm{infect}}=0.2$. Additionally, there is strong clinical and experimental evidence [26] that two physiological parameters vary significantly with SARS-CoV-2 infection: the thickness $M_{\mathrm{thickness}}$ and the mucociliary advection velocity $M_{\mathrm{vel}}$ of the mucus layer in the nasal passage. To our knowledge, the impact of host heterogeneity in these fundamental physiological features of nasal infection has never been explored, not just for SARS-CoV-2, but for any virus.

In light of the above data and results, for the present study, we explore the dynamic outcomes over 48 h in infectious viral load, total number of infected cells, and flux of infectious viral RNA copies out of the nasal passage. In this paper, we apply global sensitivity analysis techniques to our physiologically faithful, spatial respiratory infection model, focusing on the nasal passage as the source of initial infection from inhaled viruses and clinical test data from nasal swabs. As rationalized above, the global sensitivity analyses are applied across the four-parameter space of [shedding rate of infectious RNA copies, infected cell latency time, thickness, and clearance velocity of the nasal mucus layer] = [$r_{\mathrm{shedding}}$, $t_{\mathrm{latency}}$, $M_{\mathrm{thickness}}$, $M_{\mathrm{vel}}$]. For this study to be self-contained, we summarize the model and the methods before presenting the results.

### The Model

We summarize key model features from [10] so that the present paper is self-contained. As shown in Figure 1 from [10], and articulated in detail in [27], the nasal passage and all generations of the lower RT except the alveolar space are approximately cylindrical. In each generation, the epithelial cell surface is coated by a 7 μm thick layer of periciliary liquid (PCL) in which cilia beat. At full extension in the power stroke, cilia penetrate the PCL-mucus interface and extend into the mucus layer up to 1 μm, and the coordinated metachronal waves of cilia propel the mucus layer, “down” in the nasal passage and “up” in the lower RT, towards the esophagus to be swallowed.

We unfold this cylindrical geometry into a rectangular domain in which the *y*-*z*-plane falls on the epithelial cell surface. *x* denotes the “radial” distance into the PCL and mucus layers, with x=0 being the epithelium–PCL interface. *y* denotes the distance along the centerline axis, which is the primary direction of mucus advection by the coordinated beating of cilia, with y=0 representing the entry into the nasal passage. *z* is the azimuthal axis of the cylinder. Infectious virions undergo diffusion in PCL and mucus and additional advection with velocity $M_{\mathrm{vel}}$ while in the mucus layer, governed by:(1)$$\begin{array}{cc} dx & =\sqrt{2D_{v}}\;dW_{1},\\ dy & =\sqrt{2D_{v}}\;dW_{2}+M_{\mathrm{vel}}\;1_{\left\{x>{\mathrm{PCL}}_{\mathrm{gen}}\right\}}\;dt,\\ dz & =\sqrt{2D_{v}}\;dW_{3}, \end{array}$$
where
$$\begin{array}{cc} dW_{i}:\; & 1-D\;\mathrm{Brownian}\;\mathrm{motion};\\ D_{v}:\; & \mathrm{virion}\;\mathrm{diffusion}\;\mathrm{coefficient};\\ {\mathrm{PCL}}_{\mathrm{gen}}:\; & \mathrm{PCL}\;\mathrm{layer}\;\mathrm{thickness}\;(7\;\mu m\;\mathrm{uniformly}\;\mathrm{throughout}\;\mathrm{the}\;\mathrm{RT});\\ 1_{\left\{x>{\mathrm{PCL}}_{\mathrm{gen}}\right\}}:\; & \mathrm{mucus}\;\mathrm{layer}\;\mathrm{indicator}\;\mathrm{function}. \end{array}$$

Ciliated cells are the predominant infectable cells in the RT above the alveolar space, covering about 50% of the epithelial surface. Every epithelial cell has a degree of infectability, either non-infectable or with a prescribed probability $p_{\mathrm{infect}}$ of infection per encounter second.

In our model, a freely diffusing virion in the PCL encounters a cell when its distance from the epithelial cell surface vanishes, i.e., when x=0. For each second during an encounter with a ciliated cell, there is a probability $p_{\mathrm{infect}}$ of an infection. If an encounter results in infection, the cell switches from uninfected to infected, and the virion is removed from the free virion population. When the stochastic virus–cell encounter does not result in infection, for infectable or non-infectable cells, the virion is reflected back into the PCL.

Each virion is tracked until it either infects a cell or exits the generation, always toward the trachea due to strong mucus advection. Once a cell switches to an infected state, it persists in an infected, non-shedding latency state for a duration $t_{\mathrm{latency}}$, which represents cellular uptake of the virus and hijacking of the cellular machinery to replicate viral RNA copies. After $t_{\mathrm{latency}}$ has lapsed, the cell switches to a shedding state, replicating infectious virions at rate $r_{\mathrm{shedding}}$. Since infected cells typically die after 3 days post-infection, no cells switch to a death state in this 48 h study.

We assume that the kinetics of SARS-CoV-2 interactions with ciliated cells are robust within each host yet potentially highly variable between hosts, and therefore, we explore literature-supported ranges for the kinetic parameters that our previous study [22] revealed to be sensitive. All simulations to generate data for this study start at the moment of infection of one cell at the entry of the nasal passage (axial coordinate y=0). Table 1 summarizes the model parameters, fixed and variable, and the simulation details. Table 2 summarizes the three model outcomes and associated data.

## 2. Methods

We summarize previous model sensitivity analyses, their limitations, and the need for the more sophisticated, global sensitivity methods employed in the present study. In [22], we explored local sensitivity of outcomes from an initial nasal infection to host cell–virus kinetic parameters. In that study, one parameter was varied across an estimated range of possible values, while all other parameters were fixed at best-known mean estimates. While limited in scope, the following insights were gained: the total numbers of infected cells and total viral load are remarkably robust to variations in cell infectivity, $p_{\mathrm{infect}}$; shorter latency time $t_{\mathrm{latency}}$ has a dramatic, exponential effect on the progression of infected cells and total viral load; and, a higher shedding rate $r_{\mathrm{shedding}}$ of infected, post-latency cells has a significant proportional (yet non-exponential) effect on infected cell count and total viral load.

While insightful, these results suffer two important limitations that we remove in this study. First, the results correspond to one-dimensional slices in the multi-dimensional parameter space being explored and therefore lack the ability to detect if the sensitivities gained are robust to sampling off that one-dimensional slice. To generalize these limited searches of parameter space requires methods that perform global sampling and sensitivity analysis, which we summarize next and then apply. Further, the previous studies did not explore host-to-host physiological heterogeneity, which recent studies [26] have shown to arise during SARS-CoV-2 infection. We therefore add two physiological parameters, mucus thickness and advection velocity, to our global sampling and sensitivity analyses.

### 2.1. Latin Hypercube Sampling

Latin hypercube sampling (LHS) is a widely used technique to sample high-dimensional parameter spaces. It offers a quasi-random approach to efficiently sample across the entire parameter space while minimizing the number of required sampling points. Implementing LHS allows exploration of a wide range of parameters at a high resolution.

In addition, we apply partial rank correlation coefficient (PRCC) analysis to the simulated data, which we will introduce in Section 2.2. The sampling strategy of [22] contains repeated parameter values, which can impact the accuracy of PRCC results, so we cannot directly apply PRCC. Implementing LHS alleviates this issue.

LHS can be carried out as follows:Start by selecting the sample size *N*. This will be the number of our sample points in the parameter space.Determine the range and distribution of each parameter (e.g., we chose a uniform distribution for $t_{\mathrm{latency}}$ ranging from 3 h to 9 h).Divide the range of each parameter into *N* equal-probability intervals.Repeat the following steps *N* times:(a)For each parameter, randomly select one interval from the remaining pool of intervals.(b)Randomly sample from the selected intervals for all parameters.(c)Remove the selected intervals from the remaining pool of intervals.

Figure 2 shows an example of using LHS with sample size N=20 on two parameters (latency time and advection velocity with uniform distributions). We see that the range of each parameter is evenly divided into 20 intervals. Each column and each row contains exactly one sample point.

Table 3 shows the parameter ranges and distributions chosen for this analysis.

### 2.2. Partial Rank Correlation Coefficient Analysis

The PRCC method is a sensitivity analysis technique that first measures the correlations between parameters and model outcomes, and then cross-correlations are removed [28] to give the de-correlated sensitivity of outcomes to each individual parameter in Table 3. We perform this analysis at every 12 h timestamp through the 48 h following onset of infection from a single nasal cell at the entry of the nasal passage. The implementation of PRCC starts with a rank transformation of the correlation parameters $x_{j}$ and outcomes *y*. For each index *j*, we perform linear regression on $x_{j}$ and *y* in terms of other parameters:(2)$${\hat{x}}_{j}=c_{0}^{\left(j\right)}+\sum_{p=1,\;p\neq j}^{4}c_{p}^{\left(j\right)}x_{p},\;\;\mathrm{and}\;\;{\hat{y}}^{\left(j\right)}=b_{0}^{\left(j\right)}+\sum_{p=1,\;p\neq j}^{4}b_{p}^{\left(j\right)}x_{p}.$$

The PRCC is the Pearson correlation coefficient (PCC) between the residuals, $x_{j}-{\hat{x}}_{j}$ and $y-{\hat{y}}^{\left(j\right)}$, given by:(3)$$r_{x_{j}-{\hat{x}}_{j},y-{\hat{y}}^{\left(j\right)}}=\frac{\mathrm{Cov}(x_{j}-{\hat{x}}_{j},y-{\hat{y}}^{\left(j\right)})}{\sqrt{\mathrm{Var}\left(x_{j}-{\hat{x}}_{j}\right)\mathrm{Var}(y-{\hat{y}}^{\left(j\right)}})},$$
where $\mathrm{Cov}(x_{j}-{\hat{x}}_{j},y-{\hat{y}}^{\left(j\right)})$ represents the covariance between the residuals, and $\mathrm{Var}(x_{j}-{\hat{x}}_{j})$ and $\mathrm{Var}(y-{\hat{y}}^{\left(j\right)})$ represent the variance of $x_{j}-{\hat{x}}_{j}$ and $y-{\hat{y}}^{\left(j\right)}$, respectively.

The resulting PRCC value for each parameter is a number between −1 and 1, where the sign indicates positive or negative correlation and the magnitude indicates the degree of sensitivity of the outcome in question to variations in the parameter.

### 2.3. Model Simulations and Data Generation

Prior to the sensitivity analysis step, we sampled the four-parameter space using LHS as described in Section 2.1 with sample size N=20. This sample size was tested to confirm robust results. As shown in Table 4, we fix the value $p_{\mathrm{infect}}=0.2$ based on the results in [22] showing extreme robustness in outcomes over a decade or more variations. We also fix the percentage of infectable cells at 50% corresponding to the percentage of ciliated cells; this value could be slightly higher, but again, the outcomes are robust to variations [22]. We record the total shed infectious viral load, the infected cell count, and the viral flux from the nasal passage at 12, 24, 36, and 48 h post-infection of a single cell at the entry of the nasal passage.

## 3. Results

We begin by working out a specific example in detail, in which we compare PCC, Spearman correlation coefficient (SCC), and PRCC. Then we report PRCC results over all outcomes, parameters, and timestamps. We provide quantitative details because this analysis has not been previously performed on our nasal infection model.

### 3.1. Comparison of PCC, SCC, and PRCC

We use the infected cell count at 36 h as an example to demonstrate how we compute the PRCC analysis applied to simulation data from our spatial nasal infection model. Recall that we have previously established the following notations to represent the parameters, and we choose *y* to denote the outcomes.
$$\begin{array}{cc} t_{\mathrm{latency}} & \;\mathrm{latency}\;\mathrm{time}\;(\mathrm{in}\;\mathrm{hours})\\ r_{\mathrm{shedding}} & \;\mathrm{shedding}\;\mathrm{rate}\;(\mathrm{in}\;\mathrm{infectious}\;\mathrm{RNA}\;\mathrm{copies}/\mathrm{day})\\ M_{\mathrm{vel}} & \;\mathrm{mucus}\;\mathrm{advection}\;(\mathrm{in}\;\mu m/s)\\ M_{\mathrm{thickness}} & \;\mathrm{mucus}\;\mathrm{thickness}\;(\mathrm{in}\;\mu m)\\ y & \;\mathrm{outcome}\;(\mathrm{infected}\;\mathrm{cell}\;\mathrm{count}\;\mathrm{at}\;36\;h) \end{array}$$

The parameter columns in Table 5a show all 20 points in the four-dimensional parameter space selected by the LHS process. Each row represents a parameter combination and the corresponding mean simulation outcome. Figure 3 shows scatter plots of the outcome values versus each parameter.

Within each column of Table 5a, we rank-transform the column by assigning integers from 1 to 20 to values ranking from the smallest to the largest. Table 5b shows the rank-transformed parameters and outcome values. Figure 4 shows scatter plots of the ranks of outcome values versus the ranks of each parameter.

Then, we perform linear regression on each rank-transformed parameter and outcome in terms of the other parameters.
(4)$$\begin{array}{cc} {\hat{t}}_{\mathrm{latency}} & =-0.32818\;r_{\mathrm{shedding}}-0.18319\;M_{\mathrm{vel}}+0.02252\;M_{\mathrm{thickness}}+15.63295\\ {\hat{y}}_{t_{\mathrm{latency}}} & =0.90174\;r_{\mathrm{shedding}}-0.04791\;M_{\mathrm{vel}}+0.02905\;M_{\mathrm{thickness}}+1.22976 \end{array}$$
(5)$$\begin{array}{cc} {\hat{r}}_{\mathrm{shedding}} & =-0.31892\;t_{\mathrm{latency}}-0.23416\;M_{\mathrm{vel}}-0.12149\;M_{\mathrm{thickness}}+17.58306\\ {\hat{y}}_{r_{\mathrm{shedding}}} & =-0.67570\;t_{\mathrm{latency}}-0.39516\;M_{\mathrm{vel}}-0.05346\;M_{\mathrm{thickness}}+21.36043 \end{array}$$
(6)$$\begin{array}{cc} {\hat{M}}_{\mathrm{vel}} & =-0.18850\;t_{\mathrm{latency}}-0.24794\;r_{\mathrm{shedding}}-0.18314\;M_{\mathrm{thickness}}+17.00561\\ {\hat{y}}_{M_{\mathrm{vel}}} & =-0.40949\;t_{\mathrm{latency}}+0.79105\;r_{\mathrm{shedding}}+0.06213\;M_{\mathrm{thickness}}+5.84125 \end{array}$$
(7)$$\begin{array}{cc} {\hat{M}}_{\mathrm{thickness}} & =0.02442\;t_{\mathrm{latency}}-0.13555\;r_{\mathrm{shedding}}-0.19298\;M_{\mathrm{vel}}+13.69313\\ {\hat{y}}_{M_{\mathrm{thickness}}} & =-0.43254\;t_{\mathrm{latency}}+0.75422\;r_{\mathrm{shedding}}-0.13481\;M_{\mathrm{vel}}+8.53789 \end{array}$$

Finally, for each parameter $x\in \{t_{\mathrm{latency}},r_{\mathrm{shedding}},M_{\mathrm{vel}},M_{\mathrm{thickness}}\}$, we compute the PCC between the residuals $x-\hat{x}$ and $y-{\hat{y}}_{x}$ using the formula in Equation (Equation 3). The resulting numbers are the PRCCs between the parameters and the outcomes. Figure 5 shows scatter plots of the residuals of rank-transformed outcome values vs. residuals of rank-transformed parameters.

Table 6 shows a comparison between PCC, SCC, and PRCC for each parameter. Note that SCCs are obtained by computing the PCC after rank transforming the data (as in Table 5b and Figure 4), while PRCCs are obtained by computing the PCC after rank-transforming the data and taking the residuals of the data (as in Figure 5).

Scatter plots similar to Figure 3, Figure 4 and Figure 5 showing (1) raw outcome data vs. parameter values, (2) ranks of outcome data vs. ranks of parameter values, and (3) residuals of outcome data vs. residuals of parameter values for all three outcomes (total viral load, infected cell count, and flux) at 12 h time increments (12, 24, 36, 48 h) can be found in Appendix B.

### 3.2. PRCC Results

In the implementation, we use the R function epi.prcc() from the epiR package to compute PRCC between each parameter for each of the the three types of outcome data (shed infectious virion count, infected cell count, infectious virion flux via mucus clearance) at 12, 24, 36, and 48 h following the initial nasal cell infection at the entry of the nasal passage.

We observe that latency time $t_{\mathrm{latency}}$ and extracellular shedding rate of virions $r_{\mathrm{shedding}}$ have a significant impact on all infection outcomes at all timestamps. The influence of mucus advection velocity $M_{\mathrm{vel}}$ progressively intensifies from weak to somewhat strong for the total shed viral load and infected cell count as time progresses over the first 48 h post initial cell infection. Mucus thickness $M_{\mathrm{thickness}}$ within these physiological bounds has a relatively minor impact on all infection outcomes.

Figure 6 and Table 7 show PRCC results for *total viral load* for all four parameters at four 12 h time increments over 48 h post-infection. With extremely high likelihood, independent of other parameter choices, lower values of $t_{\mathrm{latency}}$ within the 3–9 h range exponentially increase total viral load at all timestamps. Similarly increasing $r_{\mathrm{shedding}}$ over a logarithmic range of 10 to 1000 infectious virions per day induces an exponential increase in total viral load at all timestamps. Slower mucus advection, as reported in [26] for COVID-19 infection, amplifies the total viral load and the number of infected cells (shown in Figure 7 and Table 8), with the effect becoming stronger over the 48 h post-infection. We do not detect a significant effect of mucus thickness.

Figure 7 and Table 8 show PRCC results for *infected cell count* at 12 h time increments for the selected parameters. The results look very similar to those in Figure 6 and Table 7, except that the effect of $M_{\mathrm{vel}}$ has a weaker time dependence, with the effect being more noticeable earlier in the infection compared to its effect on total viral load.

Note that in Table 8, the values in column “24 h” match up exactly to those in the column “36 h”. Investigation of the data shows that the ranks of the infected cell counts were preserved from 24 h to 36 h, while the raw data values changed over time. The identical PRCC values are a consequence of the identical ranks.

Figure 8 and Table 9 show PRCC results for *viral flux* (the total number of virions transported out of the nasal passage via mucus advection). Similar to previous results, $t_{\mathrm{latency}}$ has a strong negative correlation with flux and $r_{\mathrm{shedding}}$ has a strong positive correlation with flux.

Intriguingly, mucus advection velocity starts with a relatively strong positive impact on flux, but we do not detect a significant effect at later time points. We surmise this behavior is a result of the non-monotonicity of the relationship between mucus advection velocity and flux.

Figure 9 and Table 10 show the *virion flux outcomes* at various $M_{\mathrm{vel}}$ and $M_{\mathrm{thickness}}$ values while fixing $t_{\mathrm{latency}}=3$ h and $r_{\mathrm{shedding}}=100$. We see that given any fixed mucus thickness between 12.75 μm and 21.25 μm, the flux outcome values increase and then decrease as advection velocity increases from 36.67 to 220.00 μm/s. This result confirms that flux is not linearly dependent on $M_{\mathrm{vel}}$. Hence, the PRCC method cannot extract valid information about linear dependency between them.

Contour plots showing total viral load, infected cell count, and flux at various $M_{\mathrm{vel}}$ and $M_{\mathrm{thickness}}$ values at 12, 24, 36, 48 h can be found in Appendix C.

## 4. Concluding Remarks

The goal of this study is to use computational modeling to gain insights into the potential drivers of extreme population heterogeneity in SARS-CoV-2 viral titers from positive nasal swab tests throughout the pandemic. In the above sections, we summarized our physiologically faithful, spatially resolved computational model of viral infection in the human nasal passage [10]. We then described the global parameter sensitivity analyses required to evaluate the absolute and relative impact of each of four hypothesized mechanistic drivers of extreme host-to-host heterogeneity in nasal titers: nasal mucus layer thickness and clearance velocity, infected cell latency time (from the moment of infection to the onset of shedding infectious viral RNA copies) and shedding rate of infectious RNA copies. We then applied the global sensitivity methods to the model-generated, virtual population database of the dynamic progression over 48 h after initial infection of viral load, infected cells, and flux of viruses out of the nasal passage. In this virtual population, each fixed, distinct set of four parameters defines a class of similar hosts. These global sensitivity methods isolate the impact unique to each parameter, de-correlated from the other parameters, and accomplish this via quasi-random sampling over the entire four-dimensional virtual population database.

These methods produce several insightful predictions. 1. The latency time ($t_{\mathrm{latency}}$) of newly infected cells has the strongest, indeed exponential, negative correlation on total nasal viral load; i.e., linear reductions in infected cell latency time (within 9 to 3 h) produce exponential variations in total shed viral load at each 12 h timestamp, corresponding to several-orders-of-magnitude heterogeneity in viral load due solely to reduced latency time. Reduced latency time has a similar exponential impact on total infected nasal passage cell counts. 2. The viral RNA shedding rate ($r_{\mathrm{shedding}}$) of infected cells in the shedding phase has a strong, proportional but not exponential, positive correlation on total viral load at each 12 h timestamp. Orders-of-magnitude increase in shedding rate produce orders-of-magnitude increase in total nasal viral load and infected cell count. 3. Nasal mucus clearance velocity ($M_{\mathrm{vel}}$) is negatively correlated with total viral load and infected cell count, with very weak impact in the immediate hours post-infection that increases through 48 h yet mildly relative to latency time and shedding rate. 4. Nasal mucus thickness ($M_{\mathrm{thickness}}$) has little impact on infection outcomes.

The salient insight gained from this study is that the observed population heterogeneity in the first two days post nasal infection from inhaled exposure to SARS-CoV-2 can be reproduced by the mechanisms and physiological features within our computational model. This rules out other additional drivers of heterogeneity that are not captured within our model. However, this modeling and global sensitivity analysis clearly points to the latency time of infected cells—spanning cellular uptake of the virus and the hijacking of cellular machinery to produce viral RNA copies until the initial onset of extracellular shedding of viral RNA—as the primary driver of exponential population heterogeneity. Variations in the latency time of infected host cells potentially arise from some combination of viral RNA and cell DNA compatibility; e.g., there could be nuanced population DNA interactions to a specific SARS-CoV-2 variant or within variants. With respect to other respiratory viruses, the model and sensitivity results presented apply to any virus. However, to do so, one needs to have measurements of the virus–host kinetic interactions: the probability of infection per virus–cell encounter, latency time of infected cells prior to shedding of viral RNA copies, and shedding rate of viral RNA copies. These kinetic parameters are almost surely specific to virus and host, requiring cultures from the individual and exposure to the virus. This experimental data, coupled with the physiology of the individual, are predictive of pre-immune response in the immediate 48 h post initial nasal cell infection. Should features not incorporated into our modeling platform be shown to have a leading order effect, we are poised to incorporate those features, similar to how we have extended our pre-immune modeling platform to both innate [11] and adaptive [12] immunity.

These results and insights strongly suggest the need for experimental data to be collected spanning different variants of SARS-CoV-2, spanning nasal cultures grown from a diverse collection of individuals, and then careful measurements of the mechanistic parameters in our model. We note that high-resolution cell culture experiments need to focus on measurements of infection probability per virus–cell encounter, latency time, and extracellular shedding rate once an infectious virus–cell encounter takes place. The outcome metrics of total shed viral load and number of infected cells in a cell culture will not be representative of in vivo nasal infection since there is no mucus clearance in cell cultures that we know accelerates viral load. In order for these insights to be “actionable” for medical treatment, a nasal culture can determine the virus–cell infection kinetics of an individual, and single-cell measurements of latency time and replication rate could potentially guide the decision for rapid drug or antiviral therapies applied directly to the nasal passage. Lastly, the flexibility and robustness of our model and simulation platform are adaptable for future investigations into other respiratory viruses.

## Figures and Tables

**Figure 1 viruses-16-00069-f001:**
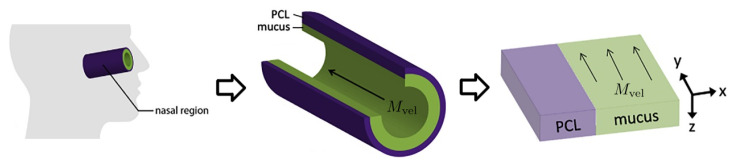
Modeling the nasal passage (image taken from [10]).

**Figure 2 viruses-16-00069-f002:**
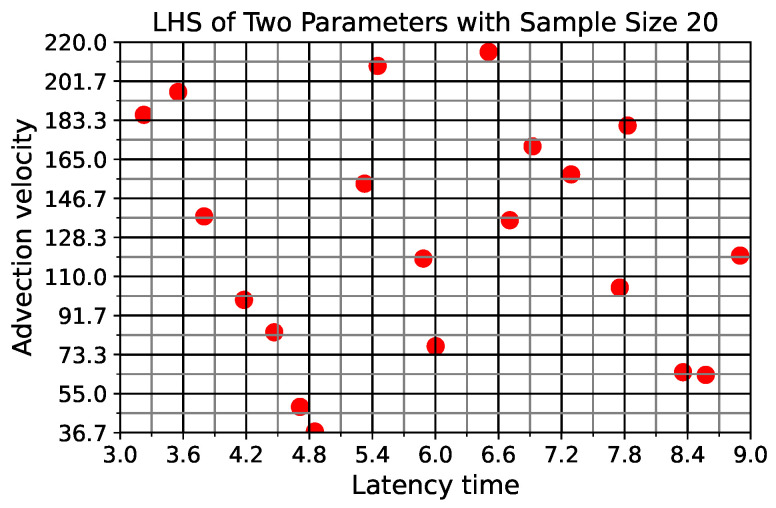
An example in which LHS is applied on two parameters using sample size 20.

**Figure 3 viruses-16-00069-f003:**
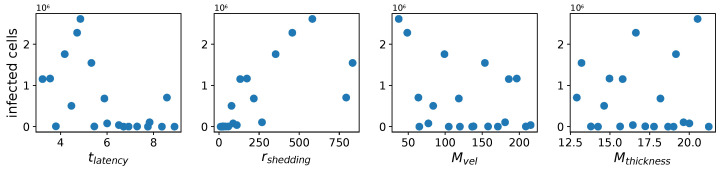
Infected cell count at 36 h vs. each parameter.

**Figure 4 viruses-16-00069-f004:**
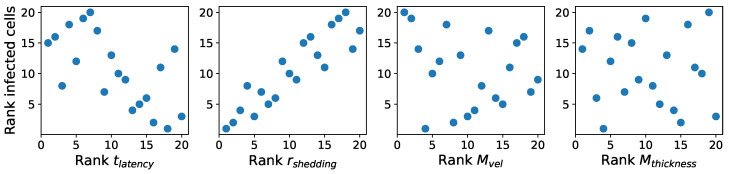
Ranks of infected cell count at 36 h vs. ranks of each parameter. Integers from 1 to 20 are assigned to values ranking from the smallest to the largest.

**Figure 5 viruses-16-00069-f005:**
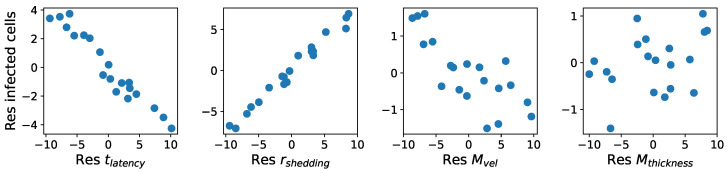
Residuals of infected cell count at 36 h vs. the residuals of each parameter. The residuals are produced by subtracting linear regression models from the outcome ranks and the parameter ranks. The CC between the residuals is the PRCC.

**Figure 6 viruses-16-00069-f006:**
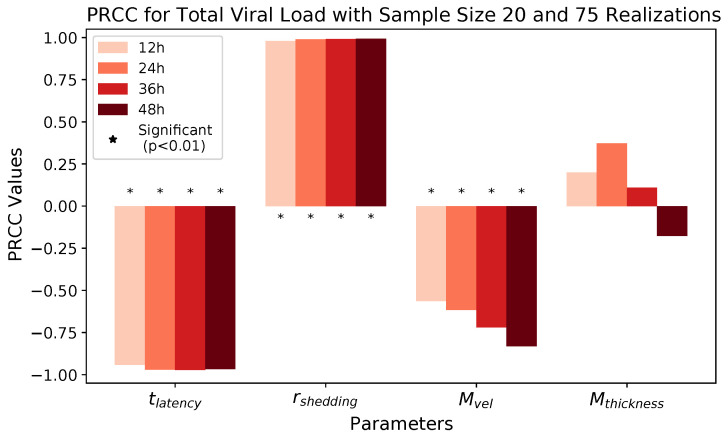
PRCC results for total viral load at 12, 24, 36, and 48 h.

**Figure 7 viruses-16-00069-f007:**
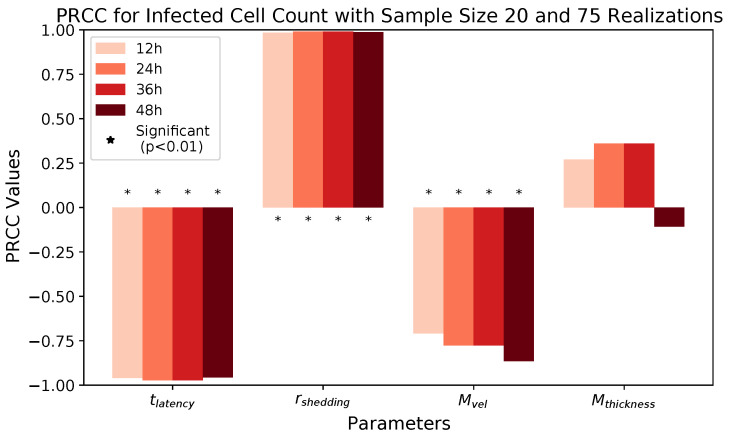
PRCC results for infected cell count at 12, 24, 36, and 48 h.

**Figure 8 viruses-16-00069-f008:**
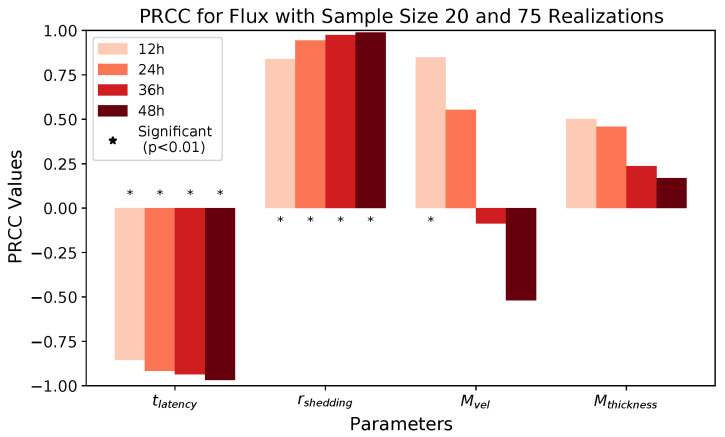
PRCC results for flux at 12, 24, 36, and 48 h.

**Figure 9 viruses-16-00069-f009:**
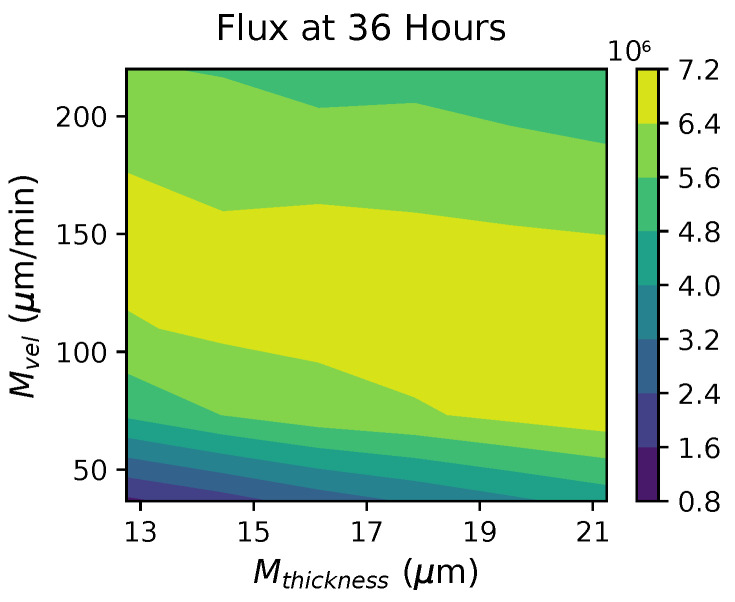
A contour plot showing the 36 h virion flux outcome for various values of mucus advection velocity and mucus thickness, while fixing $t_{\mathrm{latency}}$ at 3 h and $r_{\mathrm{shedding}}=100$ infectious RNA copies per day.

**Table 1 viruses-16-00069-t001:** The model parameters and their descriptions.

Parameters	Description
Percentage of infectable cells	The percentage of epithelial cells that are infectable
Infection probability, $p_{\mathrm{infect}}$	Probability of infection per virion–cell encounter second with infectable cells
Latency time, $t_{\mathrm{latency}}$	The time interval between the positive infectious virion–cell encounter and the onset of extracellular virion shedding
Shedding rate, $r_{\mathrm{shedding}}$	The rate (infectious virions/day) at which infected cells shed infectious viral RNA copies while in the shedding state
Mucus thickness, $M_{\mathrm{thickness}}$	Thickness of the mucus layer in the host nasal passage (the physiological mean thickness is $M_{\mathrm{mean}\;\mathrm{thickness}}=$ 17 μm)
Advection velocity, $M_{\mathrm{vel}}$	Mucus advection velocity toward the esophagus in the host nasal passage (the physiological mean nasal mucus velocity is $M_{\mathrm{mean}\;\mathrm{velocity}}=$ 146.67 μm/s)
Simulation time	Total simulation time for each model realization using fixed values for all parameters
Number of realizations	The number of realizations for each set of fixed parameters in order to obtain a robust statistical distribution of model outcomes

**Table 2 viruses-16-00069-t002:** The model outcomes and their descriptions.

Outcome Data	Description
Total viral load	The number of freely diffusing infectious virions in the nasal passage
Infected cell count	The total number of cells that have been infected
Virion flux	The number of infectious virions that have exited the nasal passage via mucus transport

**Table 3 viruses-16-00069-t003:** The 4-dimensional parameter space for the PRCC sensitivity analysis. For mucus advection velocity and mucus thickness in the nasal passage, we apply a range of multiplicative factors to the population mean average values from [10]. For mucus thickness, we choose the range to be 0.75 to 1.25 times the physiological mean thickness of 17μm. For mucus advection velocity, we choose the range to be 0.25 to 1.25 times the physiological mean nasal mucus velocity of 146.67μm/s.

Parameter	Distribution	Lower Bound	Upper Bound
Latency time, $t_{\mathrm{latency}}$	uniform	3 h	9 h
Shedding rate, $r_{\mathrm{shedding}}$	log-uniform	10 per day	1000 per day
Advection velocity, $M_{\mathrm{vel}}$	uniform	36.67 μm/s	220.00 μm/s
Mucus thickness, $M_{\mathrm{thickness}}$	uniform	12.75 μm	21.25 μm

**Table 4 viruses-16-00069-t004:** These parameter values are fixed for this study.

Parameters	Values
Infection probability, $p_{\mathrm{infect}}$	0.2
Percentage of infectable cells	50%
Simulated time	48 h
Number of realizations	75

**Table 5 viruses-16-00069-t005:** Parameter values and example outcome data: (a) shows all 20 parameter combinations selected by the LHS process, corresponding to the infected cell count at 36 h; (b) shows the rank transformed parameter values and the outcome data from a.

(a)					(b)				
**Parameters**				**Outcomes**	**Parameters** **(Ranks)**				**Outcomes** **(Ranks)**
$t_{\mathrm{latency}}$	$r_{\mathrm{shedding}}$	$M_{\mathrm{vel}}$	$M_{\mathrm{thickness}}$	y	$t_{\mathrm{latency}}$	$r_{\mathrm{shedding}}$	$M_{\mathrm{vel}}$	$M_{\mathrm{thickness}}$	y
6.71	17	136.41	18.66	269	13	3	11	14	4
5.88	217	118.40	18.18	683,012	10	14	9	13	13
6.51	111	215.41	16.44	38,445	12	11	20	9	9
3.55	174	196.61	14.97	1,167,785	2	13	18	6	16
5.33	833	153.57	13.21	1,546,895	8	20	13	2	17
8.36	10	65.06	14.24	62	18	1	4	4	1
6.92	41	171.09	17.77	1759	14	7	15	12	5
8.90	25	119.83	21.22	200	20	5	10	20	3
3.23	132	185.87	15.80	1,154,757	1	12	17	8	15
4.47	78	83.81	14.63	505,501	5	9	6	5	12
7.75	14	104.79	19.00	123	16	2	8	15	2
4.71	457	48.80	16.63	2,280,089	6	17	2	10	19
6.00	88	77.26	20.00	79,233	11	10	5	18	10
4.18	353	99.00	19.16	1,760,488	4	16	7	16	18
5.45	39	208.82	15.64	4642	9	6	19	7	7
4.85	582	37.22	20.52	2,615,026	7	18	1	19	20
8.57	793	63.81	12.91	706,248	19	19	3	1	14
3.80	24	138.19	17.22	8444	3	4	12	11	8
7.29	58	157.90	13.80	4426	15	8	14	3	6
7.83	268	180.84	19.64	105,808	17	15	16	17	11

**Table 6 viruses-16-00069-t006:** Comparisons of PCC, SCC, and PRCC for infected cell count at 36 h for all parameters.

Parameters	PCC	SCC	PRCC
Latency time, $t_{\mathrm{latency}}$	−0.54573	−0.64060	−0.97203
Shedding rate, $r_{\mathrm{shedding}}$	0.69960	0.90677	0.99036
Advection velocity, $M_{\mathrm{vel}}$	−0.40211	−0.20752	−0.77671
Mucus thickness, $M_{\mathrm{thickness}}$	−0.00101	−0.6165	0.35999

**Table 7 viruses-16-00069-t007:** PRCC results for total viral load at 12, 24, 36, and 48 h for model parameters latency time ($t_{\mathrm{latency}}$), extracellular shedding rate of infectious RNA copies ($r_{\mathrm{shedding}}$), mucus advection velocity, and mucus thickness.

	12 h	24 h	36 h	48 h
Latency time, $t_{\mathrm{latency}}$	−0.94123	−0.97044	−0.97263	−0.96750
Shedding rate, $r_{\mathrm{shedding}}$	0.97881	0.99009	0.99181	0.99327
Advection velocity, $M_{\mathrm{vel}}$	−0.56282	−0.61626	−0.71987	−0.83138
Mucus thickness, $M_{\mathrm{thickness}}$	0.19907	0.37230	0.10946	−0.17701

**Table 8 viruses-16-00069-t008:** PRCC results for infected cell count at 12, 24, 36, and 48 h for model parameters latency time ($t_{\mathrm{latency}}$), extracellular shedding rate of infectious RNA copies ($r_{\mathrm{shedding}}$), mucus advection velocity, and mucus thickness. Identical PRCCs can occur due to identical ranks.

	12 h	24 h	36 h	48 h
Latency time, $t_{\mathrm{latency}}$	−0.96017	−0.97203	−0.97203	−0.95666
Shedding rate, $r_{\mathrm{shedding}}$	0.98461	0.99036	0.99036	0.98748
Advection velocity, $M_{\mathrm{vel}}$	−0.70904	−0.77671	−0.77671	−0.86441
Mucus thickness, $M_{\mathrm{thickness}}$	0.27081	0.35999	0.35999	−0.10695

**Table 9 viruses-16-00069-t009:** PRCC results for virion flux at 12, 24, 36, and 48 h for model parameters latency time ($t_{\mathrm{latency}}$), extracellular shedding rate of infectious RNA copies ($r_{\mathrm{shedding}}$), mucus advection velocity, and mucus thickness.

	12 h	24 h	36 h	48 h
Latency time, $t_{\mathrm{latency}}$	−0.85576	−0.91655	−0.93570	−0.96692
Shedding rate, $r_{\mathrm{shedding}}$	0.83963	0.94356	0.97463	0.98938
Advection velocity, $M_{\mathrm{vel}}$	0.84918	0.55265	−0.08687	−0.51821
Mucus thickness, $M_{\mathrm{thickness}}$	0.50251	0.45874	0.23693	0.16992

**Table 10 viruses-16-00069-t010:** Virion flux values (in millions) at 36 h versus advection velocity and mucus thickness, while fixing $t_{\mathrm{latency}}=3h$ and $r_{\mathrm{shedding}}=100$ infectious RNA copies per day.

Flux ($\times 10^{6}$)		$M_{\mathrm{thickness}}$ (μm)					
		12.75	14.45	16.15	17.85	19.55	21.25
**$M_{\mathrm{vel}}$ (μm/s**)	220.00	5.66	5.54	5.28	5.28	5.04	4.78
	183.33	6.32	6.17	5.98	6.10	5.89	5.72
	146.67	6.70	6.52	6.72	6.55	6.52	6.45
	110.00	6.32	6.57	6.62	6.83	7.11	7.14
	73.33	4.92	5.61	6.07	6.29	6.62	6.90
	36.67	1.42	2.02	2.74	3.29	3.81	4.31

## Data Availability

All data are available upon request to the senior author, M.G.F.

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
