# Peer review of "Computational Modeling Insights into Extreme Heterogeneity in COVID-19 Nasal Swab Data"

_viruses, 2023, doi:10.3390/v16010069_

Round 1
Reviewer 1 Report
Comments and Suggestions for Authors
The study discusses a scientific research study that aims to understand the drivers of infection outcome sensitivity in individuals infected with SARS-CoV-2, the virus responsible for COVID-19. By applying global sensitivity methods, the researchers aim to identify and rank the drivers of outcome sensitivity during early infection. This information can provide insights into the diverse outcomes observed in the population during the COVID-19 pandemic. Overall, I recommend accepting this article after MINOR REVISIONS.
1. The research aims to investigate the potential drivers of infection outcome sensitivity in relation to physiological and host-variant heterogeneity in the context of SARS-CoV-2 infection. By applying global sensitivity methods to a physiologically faithful, stochastic, spatial model, authors aim to gain insights into the diverse outcomes observed during the COVID-19 pandemic. Here are some key points to consider in the research: Justify the assumption of suppressed immune protection based on clinical evidence of rapid nasal infections occurring largely independent of immune status; Describe how these methods allow for the assessment of the de-correlated outcome sensitivities to each source of within-host heterogeneity; Emphasize the broader applicability of your model and methods to any inhaled virus within the immediate 48 hours post-infection. Discuss how the insights gained from studying SARS-CoV-2 can contribute to our understanding of other respiratory infections.
2. “In particular, clinical nasal swab titers of non-hospitalized infected individuals have varied over the pandemic by 6 orders of magnitude [1–9].” Please provide more details.
3. Check the abbreviations throughout the manuscript and introduce the abbreviation when the full word appears the first time.
4. Some Figures in the main text are NOT important to the article, please put them in the supplementary material.
5. Present the results of the global sensitivity analysis, highlighting the dynamic rank-ordering of the drivers of outcome sensitivity at different time points (12, 24, 36, and 48 hours post-infection). Discuss the implications of these findings in understanding the population-scale outcome diversity observed during the COVID-19 pandemic.
6. Consider discussing the limitations of your study and potential future directions for research.
Comments on the Quality of English Language
Minor editing
Author Response
Thank you for your valuable feedback! Below are our responses:
1. The research aims to investigate the potential drivers of infection outcome sensitivity in relation to physiological and host-variant heterogeneity in the context of SARS-CoV-2 infection. By applying global sensitivity methods to a physiologically faithful, stochastic, spatial model, authors aim to gain insights into the diverse outcomes observed during the COVID-19 pandemic. Here are some key points to consider in the research: Justify the assumption of suppressed immune protection based on clinical evidence of rapid nasal infections occurring largely independent of immune status; Describe how these methods allow for the assessment of the de-correlated outcome sensitivities to each source of within-host heterogeneity; Emphasize the broader applicability of your model and methods to any inhaled virus within the immediate 48 hours post-infection. Discuss how the insights gained from studying SARS-CoV-2 can contribute to our understanding of other respiratory infections.
These points are addressed in the introduction and the conclusion.
2. “In particular, clinical nasal swab titers of non-hospitalized infected individuals have varied over the pandemic by 6 orders of magnitude [1–9].” Please provide more details.
The quote is a statement of fact over 3+ years, and the references are the documentation of this fact. We are not sure what more details the reviewer is referring to.
3. Check the abbreviations throughout the manuscript and introduce the abbreviation when the full word appears the first time.
We double checked and added a list of abbreviations.
4. Some Figures in the main text are NOT important to the article, please put them in the supplementary material.
We are unsure how the reviewer parsed important vs. supplementary figures. We made those choices ourselves in the best interests of the readership.
5. Present the results of the global sensitivity analysis, highlighting the dynamic rank-ordering of the drivers of outcome sensitivity at different time points (12, 24, 36, and 48 hours post-infection). Discuss the implications of these findings in understanding the population-scale outcome diversity observed during the COVID-19 pandemic.
The first point is the purpose of the paper, and we did precisely this. The second point was addressed in the original submission, but amplified in the resubmission.
6. Consider discussing the limitations of your study and potential future directions for research.
We have amplified our discussion of these topics in the Conclusion.
Reviewer 2 Report
Comments and Suggestions for Authors
Please find the attached document with comments and feedback.

Please find the attached document with comments and feedback.
Author Response
Thank for your valuable feedback! Our response is attached.

Reviewer 3 Report
Comments and Suggestions for Authors
Zhang et al. studied the global sensitivity of a physiological model of SARS-CoV-2 infection in the nasal passage within the first 48 hours. The main conclusion is that variations in physiological parameters drive the population level difference in measured viral load. In particular, the latency and shedding parameters have the most impact on the measured outcomes. While the conclusions appear reasonable, there are three main issues with the manuscript: the presentation, the motivation and support for biological conclusions, and the technical aspects. Below are my specific comments:
The presentation:
As this is a journal caters for a biological audience, the manuscript needs to be re-written in a way understandable to biologists.
1. Please add line number.
2. Starting with the abstract, it focuses on the details of the methods and not the biological results. For example, the conclusion statement makes little biological sense (“The results reveal a dynamic … during the COVID-19 pandemic”).
3. There is a significant overlap in text between different sections and unnecessary repetitions of information/figures, which shows a lack of conciseness in the writing. For example, the second paragraph in the Introduction and the first paragraph in the Methods. The first paragraph of the conclusion also simply summarizes the methods. Figures 3-5 and 6-8 could be simplified and combined. Tables 1 and 2, and tables 3 and 4 could be combined with units, range of parameters, and references.
4. The introduction reads similar to a method. The motivation is quite limited to the group’s previous work, which makes it difficult to see where the current work fits in the larger literature. Additionally, did the authors obtain permission from the journal for the reproduction of Figure 1?
5. While the authors include substantial details of the model, this is neither necessary nor sufficient. First, the model details can be delegated to previous studies – as it is not the main point of the current manuscript and readers, who want to reproduce the results, would likely look up the group’s previous studies. However, if the authors choose to do so, then sufficient information/intuition of the model needs to be included. This is where I think the model details are lacking. For example, I could not figure out what the model actually is (PDE, agent-based, network, or a hybrid?), how it is simulated, how the infection is initiated, etc., based on the current description.
6. The methods section focuses on the steps of the LHS-PRCC. The authors also give an example to demonstrate the algorithm. If the authors think this part is necessary, I suggest using a toy model with 2 parameters to demonstrate the algorithm. The main issue here is how to interpret the results biologically (e.g., what are the biological meanings behind Figures 3-5, 6-8, 9 or Tables 6-9? How should a biologist read those results?). What are the points of including PCC, SCC, and also PRCC? Do we gain new information? What are the differences biologically or otherwise? What do some of the conclusions mean? For example, what is the biological meaning of a “strong positive PRCC”, etc.?
7. If possible, it would be great to have the factors that affect the magnitude of the model parameters. For example, does age affect the mucus layer, advection velocity, etc., and if so, how? This would add further biological insights into the current study. Also, the last paragraph of the manuscript main text feels very generic and not specific to the results.
The motivation and support for biological conclusions:
In general, when making statements such as (Result, 2nd paragraph) “… have a significant impact … at all timestamps”, the authors should also include the supporting results (figures/tables).
1. The question of the study is not well motivated. The authors stated that (Introduction, 1st paragraph) “… have varied over the pandemic by 6 orders of magnitude” and referred to references 1-9. This is very vague. (1) How is this difference being compared? Is this the difference of the peak viral load, or at a particular time after infection across all individuals infected with SARS-CoV-2 in those 9 studies? (2) Are the tools used to measure viral load similar across those 9 studies? (3) Are those studies focused on a single strain (which is the underlying assumption of the conclusion of the current study)? I suggest being more specific about this motivation.
2. Some conclusions are very strong and not supported by the results. Specifically, (Concluding Remarks, 4th paragraph) “… observed population heterogeneity … completely explainable from a mechanistic …” is very strong. There are numerous limitations and underlying assumptions in the current study that are not discussed by the authors. For example, most existing data are collected many days after the infection, but this study focuses on the first 48 hours. How does the simulation results from the first 48 hours apply to explain the difference elsewhere. This comment applies to the other two major conclusion in the 5th and 6th paragraphs in the Concluding Remarks. Is there any evidence in literature that helps support these conclusions?
The technical aspects:
1. The issue that the authors ran into and discussed at the end of the Results is a common issue due to an incorrect application of LHS-PRCC, namely a monotone test. I suggest including a monotone test prior to the application of LHS-PRCC for all model variables.
2. Why did the authors choose to use 20 samples? If there are 4 parameters being studied, and assume there are low, mid, or high value for each parameter, I would think at least 64 samples are needed.
Author Response
Thank you for your valuable feedback! Our response is attached.

Round 2
Reviewer 2 Report
Comments and Suggestions for Authors
Authors have addressed the my comments and concerns appropriately.